# Impact of Opioids on Cellular Metabolism: Implications for Metabolic Pathways Involved in Cancer

**DOI:** 10.3390/pharmaceutics15092225

**Published:** 2023-08-29

**Authors:** Doorsa Tarazi, Jason T. Maynes

**Affiliations:** 1Department of Biochemistry, University of Toronto, Toronto, ON M5G 1A8, Canada; doorsa.tarazi@sickkids.ca; 2Program in Molecular Medicine, The Hospital for Sick Children, Toronto, ON M5G 1X8, Canada; 3Department of Anesthesia and Pain Medicine, The Hospital for Sick Children, Toronto, ON M5G 1X8, Canada; 4Department of Anesthesiology and Pain Medicine, University of Toronto, Toronto, ON M5G 1E2, Canada

**Keywords:** opioid, cancer, metabolic pathway, morphine, metabolomics, tumor, glycolysis, TCA

## Abstract

Opioid utilization for pain management is prevalent among cancer patients. There is significant evidence describing the many effects of opioids on cancer development. Despite the pivotal role of metabolic reprogramming in facilitating cancer growth and metastasis, the specific impact of opioids on crucial oncogenic metabolic pathways remains inadequately investigated. This review provides an understanding of the current research on opioid-mediated changes to cellular metabolic pathways crucial for oncogenesis, including glycolysis, the tricarboxylic acid cycle, glutaminolysis, and oxidative phosphorylation (OXPHOS). The existing literature suggests that opioids affect energy production pathways via increasing intracellular glucose levels, increasing the production of lactic acid, and reducing ATP levels through impediment of OXPHOS. Opioids modulate pathways involved in redox balance which may allow cancer cells to overcome ROS-mediated apoptotic signaling. The majority of studies have been conducted in healthy tissue with a predominant focus on neuronal cells. To comprehensively understand the impact of opioids on metabolic pathways critical to cancer progression, research must extend beyond healthy tissue and encompass patient-derived cancer tissue, allowing for a better understanding in the context of the metabolic reprogramming already undergone by cancer cells. The current literature is limited by a lack of direct experimentation exploring opioid-induced changes to cancer metabolism as they relate to tumor growth and patient outcome.

## 1. Introduction

Opioids are widely used for pain management within the cancer patient population, for surgical and non-surgical pain treatment. Pain experienced by patients decreases quality of life and may also impact disease prognosis. Recent studies have reported that pain and subsequent opioid use were significant predictors of clinical outcomes in cancer patients [1,2,3]. Approximately 30–50% of patients undergoing antineoplastic treatment and 75–90% of patients with advanced disease experienced severe pain necessitating opioid therapy [4]. Opioids are administered through various routes including oral, intravenous, transdermal, and subcutaneous. These medications act primarily on the central nervous system by binding to opioid receptors in the brain and spinal cord involved in pain perception [5,6,7]. Binding of opioids to their cognate Class A G-protein coupled receptors in the central and peripheral nervous system results in receptor internalization and activation of multiple cellular signaling cascades [8]. This signal transduction eventually leads to the formation of descending inhibitory impulses through inhibition of GABA interneurons, ultimately decreasing nociceptive transmission and providing pain relief [7]. Despite their primary use for analgesia, administration of opioids can affect other organ systems through off-target and on-target receptor binding, including but not limited to gastrointestinal, respiratory, cardiovascular, neurological, and renal systems [9]. Newer opiate agonists, like oliceridine, have been developed to maintain analgesic efficacy but produce fewer undesirable side-effects [10,11].

There are three general types of opioids: natural, semi-synthetic, and synthetic. Morphine, a natural opiate derived from opium, is the most commonly used opioid for cancer pain [5,12,13]. Given the frequency of use during cancer treatment, tolerance to one opioid may require increasing doses or the need for a combination of drugs for effective pain relief. A comprehensive list of opioids, their classification, and their affinity for opioid receptors is in Table 1, accompanied by their chemical structures in Figure 1 [14]. Given the ubiquitous need for opioids in cancer pain management, their potential effects on tumor treatment and progression and patient prognosis has been studied [15,16,17,18,19]. A majority of this work has focused on morphine and concluded that the opioid has differential effects based on tumor classification. Despite the recognition that metabolism is a significant driving factor for tumor potentiation and growth, studies looking at the effect of opioids generally fail to address the possible impact of these drugs on cancer metabolism [20]. Opioid-induced changes to cellular metabolism are widely studied in the context of dependence, tolerance, and withdrawal. Research in this field is predominantly limited to changes observed in otherwise normal brains, urine, blood plasma and liver tissues. Given the role of metabolism in oncogenesis and the derangements induced by individual cancer subtypes, further changes to cancer-related metabolic pathways induced by analgesic use are of interest, especially for CNS-based tumors as this is the main site of action for opiates [21].

In this review, we provide an overview of changes to key cancer-specific metabolic pathways induced by opioids. Opioid-induced metabolic changes may have the potential to exacerbate or ameliorate certain pathways already affected by cancer metabolic reprogramming. Understanding these changes will better inform decisions regarding opioid prescription and usage for cancer patients, as well as the potential effects on patient outcomes [10]. While the outlined studies demonstrate a thorough examination of cellular metabolic changes, a significant gap in knowledge persists in opioids’ direct impact on cancer cell metabolism as it relates to disease pathogenicity and patient outcome. This review underscores opioid-induced alterations in crucial metabolic pathways pertinent to cancer. Given the heterogeneity of cancer, meticulous exploration of each tumor type, and possibly patient-specific investigation, is imperative. Such understanding is key to ascertain the therapeutic value or detriment of specific opioids in pain management, guided by metabolic nuances, for optimal patient outcome.

**Table 1 pharmaceutics-15-02225-t001:** Common opioids use in cancer pain management and their opioid receptor specificity.

Opioid Name	Classification	Opioid Receptor Specificity	Ref.
Morphine	Natural	MOR > DOR > KOR	[22,23]
Codeine	Natural	MOR	[24]
Oxycodone	Semi-synthetic	MOR > KOR > DOR	[25,26]
Hydromorphone	Semi-synthetic	MOR > DOR > KOR	[27,28]
Buprenorphine	Semi-synthetic	MOR	[29]
Hydrocodone	Semi-synthetic	MOR > KOR > DOR	[30]
Methadone	Synthetic	MOR > KOR > DOR	[31,32]
Fentanyl	Synthetic	MOR > KOR	[33,34]
Tramadol	Synthetic	Weak (MOR, KOR, DOR)	[35,36]

MOR—mu (μ) opioid receptor; DOR—delta (δ) opioid receptor; KOR—kappa (κ) opioid receptor.

**Figure 1 pharmaceutics-15-02225-f001:**
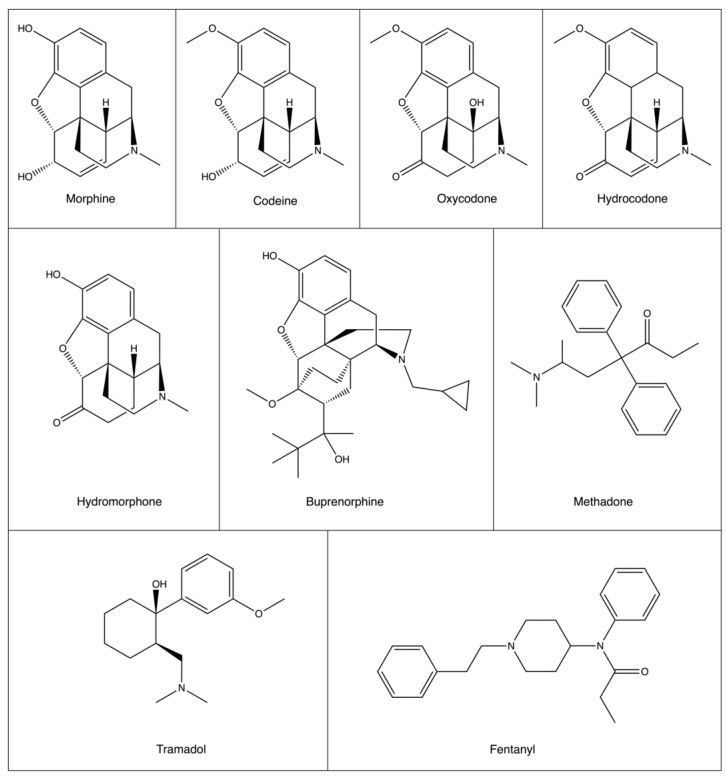
Structures of common opioids used in cancer pain therapy. While there is structural similarity between synthetic and semi-synthetic opioids, synthetic opioids methadone, tramadol, and fentanyl are structurally very different.

## 2. Modulation of Cancer Growth by Opioids

Comprehensive cohort and retrospective studies conducted on patients with different types of cancer have shed light on the complex relationship between opioid use and cancer recurrence [37,38,39,40,41,42,43]. The investigations encompassed diverse cohorts, methodologies, and treatment modalities, revealing intriguing patterns and potential therapeutic implications. Because of their retrospective nature, numerous studies amalgamate the results around analgesia and anesthesia rendering it challenging to make truly distinct conclusions around the benefits or detriments of specific opioid use. For example, a clinical retrospective study of breast cancer patients who had either received a paravertebral nerve block with general anesthesia or general anesthesia with morphine found that anesthetic and analgesic technique had a significant impact on cancer recurrence. At 36 months, patients who received a paravertebral nerve block had a 94% recurrence-free and metastasis-free survival rate, compared to only 77% for those who received opioid analgesia [44]. Numerous retrospective studies focusing on non-small cell lung cancer patients and advanced prostate cancer demonstrated that higher dosages of postoperative morphine or generally higher opioid use was significantly correlated to higher cancer recurrence in a 5-year timeframe (hazard ratio (HR) 1.127), decreased progression-free survival, and decreased overall survival (HR 1.55) [38,39,42]. When comparing recurrence-free versus tumor recurrent groups, non-small cell lung cancer patients who had experienced tumor recurrence in a 5-year time period had received two times the total doses of postoperative opioid administration [42]. Dosage of pain management drugs were provided based on patient need. In lung adenocarcinoma patients, ketamine administration was protective for recurrence-specific survival (HR 0.44) but higher opioid use was associated with worse survival (HR 1.09). However, patients who exhibited genomic alterations in the oncogenic Wnt and Hippo pathways showed a positive correlation (~10–30% improvement) between increased opioid dose and recurrence-specific survival [37]. In a 12-year cohort study investigating over 200,000 patients with breast cancer, no strong association between opioid use or dosage and cancer recurrence was seen [41]. MOR antagonism, specifically in non-neuronal tissue, has also been explored as a possible method of attenuating cancer progression, focussing on mitigation of off-target consequences of opioid use. Methylnaltrexone, a peripheral MOR antagonist that cannot cross the blood—brain barrier was initially being investigated for its laxative properties in 229 patients with lung, breast, prostate, or pancreatic cancer experiencing opioid-induced constipation (NCT00401362, NCT00672477). However, when examining the pooled data from both randomized control trials, researchers also observed longer median overall survival by 76–118 days in patients receiving methylnaltrexone [40]. These findings challenge the notion of a universal relationship between opioids and cancer progression and emphasize the need for tumor-specific analyses to elucidate the cellular mechanisms involved.

The effect of opioids on tumor growth and proliferation has been studied using in vivo and in vitro models [15,45,46,47,48,49]. Numerous comprehensive reviews have highlighted that opioids, particularly morphine, have diverse effects on cancer phenotypes [15,17,50,51]. In in vitro models of breast cancer carcinoma and lung cancers, morphine has been shown to increase apoptosis and decrease cell proliferation through MOR/KOR activation and reduction in protein kinase C (pro-survival pathway) activity. Morphine was shown to inhibit the transcriptional regulator NFkβ and reduce expression of the pro-inflammatory cytokine TNF-α [52,53]. Both molecules have been associated with increased aggression of cancer phenotypes such as proliferation, metastasis, and immune evasion [54]. Morphine-induced reduction of NFkβ and TNF-α leads to inhibition of tumor cell growth in a variety of cancer cells [52,53]. Conversely, morphine has also been shown to upregulate VEGF and stimulate VEGF receptor transactivation as a pathway for increased cell proliferation [55,56,57]. Upregulation of VEGF has been connected to promotion of glycolysis and Warburg metabolism in pancreatic cancer [58]. Morphine-induced activation of Akt and mTORC1 pathways were shown to increase proliferation, cell migration, and invasion with indications that this may be through upregulated glucose metabolism, phosphorylation of sarcoma oncogene cellular homologue (Src), and augmentation of nutrient transports [55,59,60]. Fentanyl was shown to inhibit both tumor growth and invasion by downregulating β-catenin, a protein that has numerous roles as a signal transducer of the Wnt pathway and a modulator of lipid and carbohydrate metabolic enzymes [49,61].

The majority of studies examining the impact of opioids on cancer progression focus on phenotypic alterations, including proliferation, invasion, metastasis, and apoptosis. These studies delve into the translational and transcriptional mechanisms underlying these changes, often implying a potential involvement of metabolic processes [17,50]. The reprogramming of cellular metabolism is a fundamental characteristic of cancer and is a substantive mechanism through which opioids may influence the aforementioned cancer phenotypes [62,63].

## 3. Cancer Metabolism

Early discoveries indicated that cancer cells increase glucose consumption for energy synthesis, leading to an interest in downstream handling pathways of glycolysis and the tricarboxylic acid cycle (TCA), and anaplerotic pathways such as serine and glutamine metabolism [62]. While early concepts regarding cancer initiation suggested that heightened aerobic glycolysis could potentially serve as a distinguishing feature of cancer (involving reduced mitochondrial respiration and compensatory lactate fermentation), cancer metabolism is far more nuanced. Glycolytic upregulation is specific to certain cancer types. Significant mitochondrial ATP production via oxidative phosphorylation (OXPHOS) and the electron transport chain (ETC) is often needed for sufficient energy production for the hypergrowth of tumors [64]. A consequence of increased OXPHOS is the higher production of reactive oxygen species (ROS). Increased ROS can facilitate tumorigenesis through direct DNA damage and ROS-based signaling pathways but must be appropriately balanced with detoxification pathways such as the pentose phosphate pathway (PPP) and glutathione (GSH) cycle, in order to prevent tumor cell death [65]. Through the dysregulation of this non-exhaustive list of metabolic pathways, cancer cells promote growth, evade apoptosis, and initiate metastasis.

## 4. Glycolysis

### 4.1. Glycolysis in Cancer

A distinctive metabolic trait of many cancer cells is the increased uptake of glucose as a substrate for glycolysis. Despite being less efficient than OXPHOS, the Warburg effect, first described in 1920 by Otto Warburg, proposed that cancer cells preferentially upregulate glycolysis with lactate fermentation (potentially still in the presence of oxygen, hence the term aerobic glycolysis) to synthesize ATP [66]. This metabolic shift is initiated by several factors including the activation of oncogenes KRAS and BRAF, changes in the tumor microenvironment, and the inhibition of the tumor repressors NDRG2 and p53 [67,68,69,70]. Many cancer types overexpress GLUT (glucose transport) enzymes, particularly in hypoxic conditions, to accentuate cellular glucose uptake as a rate-limiting step of glycolysis [71,72]. Interestingly, glycolytic enzymes are ubiquitously upregulated in most cancers. In brain tumors, all glycolytic enzymes, including hexokinase-1 which is stringently allosterically regulated by glucose-6-phosphate (G6P), are overexpressed [73]. Phosphoglucose isomerase (PGI), phosphofructokinase (PFK), enolase-1 (ENO), and pyruvate kinase (PK) are the most frequently overexpressed glycolytic enzymes across various cancer types (Figure 2) [74]. Glycolytic upregulation provides a quick source of ATP and additionally provides a method to shuttle glycolytic intermediates into anabolic pathways to promote the synthesis of nucleotides, amino acids, and lipids for cell proliferation and survival [75,76,77]. For these reasons, modulation of glycolytic activity in cancer cells has been an ongoing point of investigation for treatment [78].

### 4.2. Opioid Modulation of Glycolysis

Metabolic studies on the effect of morphine in rodent brain samples have indicated an increase in intracellular glucose following both acute and chronic opioid treatment [79,80,81,101]. In murine colon cancer cells, morphine increased the expression of GLUT1, providing a mechanism for increased intracellular glucose levels and suggesting that morphine could induce changes to glucose import pathways or push for increased glycolytic ATP-generation over OXPHOS [18,102]. Aldolase C, the brain-specific isoform of fructose-1,6,-bisphosphatase responsible for the production of glycerol-3-phosphate and dihydroxyacetone during glycolysis, is increased in the mouse hippocampus following chronic morphine treatment at concentrations ranging from 20 to 50 mg/kg/day. This increase in protein expression is reversible with the administration of naloxone indicating an opioid receptor-specific pathway [103]. Acute morphine treatment at high doses of 50 mg/kg for fewer than 2 h in rats increased glycogen, fructose-1,6-bisphosphate (F-1,6-BP), pyruvic acid, lactic acid, ATP, and ADP, while decreasing G6P in the brain [80]. Within the same treatment time frame, lower morphine concentrations (15 mg/kg) increased G6P, and fructose-6-phosphate, but had no effect on F-1,6-BP. This observation may indicate delicate temporal regulation in the analysis and concentration-dependent variations. [79,82]. Changes to levels of G6P in the context of cancer may be of more relevance to the PPP, as fluctuations in abundance of this metabolite are often seen as a source of glucose-6-phosphate dehydrogenase (G6PDH) upregulation and will be discussed further in Section 9. Overexpression of G6PDH is negatively associated with tumor stage, depth of invasion, lymph node metastasis and ultimately survival rate [104]. In subcutaneous primary melanomas, glycolytic metabolite levels including both G6P and F6P were elevated in metastatic tumor samples compared to their non-metastatic counterparts. Elevation of these metabolites is suggested to support metastasis by upregulating the PPP serving as a mechanism to attenuate potential apoptotic signaling induced by high levels of ROS [105]. Thus, alterations to these metabolites brought on by morphine treatment may influence metastatic potential in cancer cells.

Many of the aforementioned glycolytic changes are mediated through the opioid receptor, as blocking the receptor with nalorphine treatment returned the metabolic phenotype back to control [79]. An extensive proteomic study investigating the immunosuppressive effects of chronic morphine treatment in African green monkeys showed reduced expression of multiple glycolytic enzymes. Consistent dosing with 5 mg/kg morphine over the 20-day treatment period reduced expression of glyceraldehyde-3-phosphate dehydrogenase (GAPDH), phosphoglycerate mutase (PGM), ENO, and phosphoglycerate kinase (PGK) in varying tissue samples [83]. The activity and expression of PK following morphine exposure is contested in the literature. PK expression was elevated in neuronal tissue of rats and mice when treated for 1–5 days at concentrations of 5–15 mg/kg [84,106,107]. However, in investigations looking at non-primates and neuronal rat tissue treated with 5–50 mg/kg morphine for 10–20 days, PK expression decreased [83,85,86]. PK downregulation can be an indication of increased glycolytic flux and is a known mechanism for promoting cancer cell survival [108]. These studies may suggest that longer and higher concentrations of morphine exposure correspond to lower PK expression, while shorter and lower concentrations increase expression. While all metabolites and enzymes of the pathway are of interest for studying, changes to regulatory branches at HK, PFK, and PK are of particular importance as their modulation can significantly impact pathway flux [109].

Regulation of lactate dehydrogenase activity is important for the balance of pyruvate and lactate, and plays a role in tumorigenesis. Lactate dehydrogenase A (LDH-A) is a transcriptional target of the oncogene c-MYC and is involved in the upregulation of glycolysis in cancer cells that rely on Warburg metabolism [66,110]. Lactate dehydrogenase B (LDHB) more readily oxidizes lactate to pyruvate while the A subunit is involved with the reverse reaction as part of anerobic glycolysis. Both acute and chronic morphine administration (10–100 mg/kg) alter the expression level of components of the LDH complex, increasing LDHA but decreasing LDHB mRNA and protein expression; a change not observed with tramadol treatment [18,87,88,89,110]. The changes to LDH are thought to be a compensatory mechanism to help fulfil immediate energy demands. Along with changes at the enzymatic levels, metabolite and proton-NMR-based metabolomic analysis of rat brain and plasma following acute morphine treatment have shown an increase in lactic acid levels [81,90]. Lactic acid is produced when the energy demand of the brain exceeds the production capacity of OXPHOS leading to upregulation of glycolysis and glycolytic stress [111]. Regulation of lactic acid can significantly impact tumor growth and metabolic homeostasis. It is well-established that increased lactic acid levels act as fuel for oxidative metabolism in hypoxic tumor cells and have the potential to increase metastasis by approximately 10-fold [112,113,114]. Moreover, increased lactic acid in the tumor microenvironment allows for cancer cells to overcome immune surveillance by preventing natural killer cell activation and T-lymphocyte proliferation [115].

Expression of the components of pyruvate dehydrogenase, the connecting enzyme between glycolysis and the TCA by converting pyruvate to acetyl-CoA, are also reduced with chronic morphine treatment in mouse hippocampal tissue [87]. Changes in the expression levels of enzymes at the interface between aerobic and anerobic glycolysis, as well as the TCA, suggest morphine can alter the cell’s reliance on oxidative energy metabolism.

While some of the observed metabolic changes appear to be mediated through on-target effects (through the opioid receptor), others are independent of receptor presence or activation, indicating potential off-target binding. Fentanyl and tramadol, both of which also provide analgesia through MOR activation, do not increase lactic acid levels’ [91]. This suggests that the mechanism of morphine-induced increases in lactic acid may be independent of MOR activation. Because lactic acid levels in the tumor microenvironment can significantly impact patient prognosis, considerations should be made to choice of analgesic intervention, given that morphine can increase intercellular lactic acid levels where fentanyl treatment does not [116].

Five-week tramadol treatment in mice significantly decreases fructose-2,6-bisphospate and fructose-6-phosphate in the cerebellum [91]. Fructose-2,6-bisphosphate can modulate the activity of PFK, the second commitment step of glycolysis and a key regulatory point. A drop in levels of these two metabolites suggests an impedance of glycolysis by tramadol [117]. This finding may point to a shift in energy dependence to OXPHOS or suggest lower energy/ATP availability. Changes in metabolite levels in these studies were highly dependent on the concentration of drug used, illustrating dose-effect.

Noscapine, an opium alkaloid with limited analgesic properties, has recently displayed potent anti-tumor capabilities. A study in human renal cancer has suggested that noscapine treatment (concentration undisclosed) modulates the Warburg effect through activation of PI3K/mTOR signalling by lowering intercellular glucose uptake and both lactic acid and ATP production [118]. Despite not having analgesic use itself, noscapine can potentiate the analgesic effect of morphine three-fold when used in combination [119]. While noscapine has been investigated as an anti-tumor drug, further work is required to fully elucidate the mechanisms by which it increases apoptosis in cancer cells, and whether this holds true when combined with opioids for pain relief.

## 5. Creatine Metabolism

### 5.1. Creatine Metabolism in Cancer

Creatine kinase catalyzes the reversible reaction of phosphocreatine to creatine for ATP generation [120]. The enzyme can also act as an energy transport system to shuttle high-energy phosphate molecules to ATP utilization sites in the cytosol and mitochondria [121]. Creatine provides an important energy reserve predominantly in cells that are excitable and highly efficient—namely neurons and myocytes [122]. Historically, creatine treatment in cancer (artificially increasing creatine and phosphocreatine levels) has been associated with suppression of cell proliferation via modulation of energy buffering capacity and even increasing muscle mass in patients [123,124,125,126]. However, the role of creatine and its analogs in cancer are not straightforward. Studies in the last decade have called attention to creatine’s ability to induce cancer progression and initiate metastasis [127,128,129,130]. Increased creatine levels have been associated with poor survival in hepatocellular carcinoma, vulvar cancer, and breast cancer [131].

### 5.2. Opioid Modulation of Creatine Metabolism

Morphine can significantly alter phosphocreatine levels within acutely treated cells. Reports on the direction of change and the downstream effect on ATP production vary from study to study [79,80]. However, recent findings suggest that changes to creatine and phosphocreatine levels are dependent on specific brain structures. While chronic morphine treatment of 14 days increased creatine in the hippocampus and nucleus accumbens, it decreased in the prefrontal cortex and striatum [90]. In rhesus monkeys, chronic morphine treatment for 90 days at 10 mg/kg had the opposite effect, with increased creatine in the prefrontal cortex and a decrease in the hippocampus [92]. Increased creatine found in the brain of morphine-dependent rats could also point to a protective mechanism against oxidative stress [90,132,133]. Previous studies have suggested that elevated levels of creatine in the hippocampus are associated with improved memory and recall [134]. In a patient-derived xenograft mouse model of colon cancer, increased creatine abundance both through dietary intake and upregulation of enzymatic creatine production enhanced metastasis and reduced survival time [127]. This change is a result of downstream creatine-induced upregulation of snail and slug proteins which are well-established targets of Smad2/3 and promoters of tumor metastasis [135]. A morphine-induced creatine increase in sublocalized areas of brain tissue could significantly impact metastatic ability in the presence of tumors cells.

Morphine was shown to bind creatine kinase B at the micromolar level, inhibiting the enzyme’s ability to phosphorylate ADP and thus decreasing ATP production in rat brains [136]. Morphine treatment for only 2 days in mice decreased creatine kinase B expression in hippocampal tissue [84]. However, the overwhelming current data suggest that chronic morphine exposure reduces intercellular ATP levels [87,137,138]. High ATP production is the primary determinant of aggressive cancer phenotypes, and targeting the production of ATP has been successfully explored as cancer therapy [139,140]. Because morphine is able to reduce ATP production in cells, it may be a superior option to other opioids for cancers that are particularly sensitive to fluctuations in ATP levels [141].

## 6. Glutamine Metabolism

### 6.1. Glutamine Metabolism in Cancer

Glutamine is used as a major carbon source for cancer cells to feed into the TCA, acts as a nitrogen donor for the synthesis of purine and pyrimidines needed for DNA replication, and provides a precursor for GSH synthesis [142]. The metabolite is therefore integral for the production of energy and the maintenance of redox balance. Many tumors are so dependent on glutamine that they are categorized as “glutamine addicted” cancers, including lymphomas, glioblastomas, colorectal, and breast cancers [143]. These cancers heavily rely on glutamine for the synthesis of glutamate, α-ketoglutarate (αKG), and citrate [144]. Following entrance into the cell, glutamine is converted to glutamate through the activation of glutaminase (GLS) (Figure 2). Upregulation of GLS is associated with more advanced disease and poor prognosis of hepatocellular carcinoma, colorectal, breast, and pancreatic cancer [145,146,147,148]. These cancer types so heavily depend on glutamine such that disruption to its synthesis or levels is detrimental to tumor cell survival [144,149].

### 6.2. Opioid Modulation of Glutamine Metabolism

Glutamine can be synthesized by the enzyme glutamine synthetase (GS) through the combination of glutamate and ammonia. It can also be converted back into glutamate via GLSase as part of the glutamine–glutamate cycle, which is imperative for GABA production and neuron activation [150]. A study looking at changes to GS levels in rats following 72 h of consistent morphine treatment found that opioid exposure did not alter enzymatic activity, nor did it affect enzyme expression levels in brain tissue [107]. However, short-term treatment of under 1 h did decrease GLS protein expression [93]. NMR studies in the rat brain following chronic morphine treatment found a minor increase in both glutamate and glutamine levels in the locus coeruleus and periaqueductal gray [94]. Other studies have found glutamate to be increased in the hippocampus, nucleus accumbens, and striatum, but decreased in the prefrontal cortex of morphine dependent mice [90,95]. In rat plasma samples, both glutamine and 4-hydroxybutanoic acid, a precursor to glutamine, were decreased following morphine treatment (5–50 mg/kg), both acutely (within 24 h) and chronically at 5 days [81,96]. Using deuterium magnetic resonance spectroscopy (^2^H DMRS), the rate of cerebral glutamine consumption in morphine-treated mice was found to be approximately two-fold higher than control [151]. Moreover, a 90-day morphine addiction study performed in rhesus monkeys indicated that consistent exposure to 10 mg/kg morphine decreased both glutamate and glutamine by 7–10% in the hippocampus. Interestingly, the treatment resulted in a 7% increase in glutamine in the prefrontal cortex [92]. Morphine treatment induces differential regulation of GABA, glutamine, and glutamate in relation to brain structure location. Tramadol treatment in the cerebrum of mice resulted in increased γ-hydroxybutyric acid, a precursor to glutamine, while actual glutamine levels were found to decrease [91]. A study on neurotransmitter release rate in the hypothalamus found that rats treated with fentanyl for 75 min had slower glutamate and faster GABA release. However, no comment was made on total metabolite levels [152]. Rapidly proliferating cancer cells depend on glutamine uptake for anapleorosis into the TCA cycle to eventually produce ATP, NADH, and FADH_2_. As such, enzymes that regulate the glutaminolysis are attractive druggable targets [153]. Decreased glutamine uptake through inhibition of glutamine transporters has been shown to reduce cancer growth by preventing cell cycle progression through E2F transcription factors [154]. Similarly, depriving cancer cells of glutamine can enhance their sensitivity to apoptosis induced by TNF-α and heat shock [155]. Glutamine is also a key modulator of mTORC1 activation, a kinase which regulates cell growth and proliferation [156]. While morphine is reported to activate mTORC1 signaling independent of cellular metabolism, its ability to affect glutamine levels, metabolism, and uptake are likely systems by which it modulates cancer growth [60].

## 7. Tricarboxylic Acid (TCA) Cycle

### 7.1. The TCA Cycle in Cancer

The TCA cycle is a central hub for energy production, redox balance, and metabolite synthesis (Figure 2). Identification of mutations in isocitrate dehydrogenase, succinate dehydrogenase, fumarase, and malate dehydrogenase (MD) in a variety of cancer types has redirected attention to the role of the TCA in reprogramming tumor metabolism [157,158,159]. Increased levels of succinate were shown to enhance cell migratory and metastatic ability by activation of succinate receptor-1-mediated signaling [160]. αKG supplementation was shown to have anti-proliferative effects by attenuation of Wnt signaling and suppression of TGF-β [161,162,163]. Despite being initially overlooked as a dysregulated metabolic pathway in cancers, the TCA cycle has become a key target for cancer therapy [144,164,165,166].

### 7.2. Opioid Modulation of the TCA Cycle

The most extensive work regarding changes to TCA metabolites following opioid exposure were performed using blood and urine samples from rodents. Morphine has been reported to significantly impact numerous metabolites of the TCA cycle. In a study examining morphine relapse where exposure was increased over a 10-day period (3–9 mg/kg), followed by a 5-day period of withdrawal, then morphine relapse for up to 20 days, changes to 30 responsive metabolites in serum samples were noted. Upon initial exposure, mice had decreased levels of pyruvic acid and αKG, but no effective change was seen in citric acid. After withdrawal and relapse, a decrease in citric acid and increase in both αKG and pyruvic acid was observed [97]. Heroin was utilized as a comparative drug to morphine in these experiments. Heroin treatment for the same duration had dissimilar effects, despite being considered a prodrug that is metabolized into morphine in vivo, indicating potential off-targets for either (or both) drugs. Heroin resulted in an initial increase in citric acid, a decrease in αKG, and a decrease in pyruvic acid levels. Following withdrawal and relapse, almost all observed metabolites were decreased compared to the saline control group, except for pyruvic acid, which was found to be no different than control. Elevated pyruvic acid can inhibit cell growth through repression of histone gene expression and subsequent delay of S-phase entry [167]. Pyruvic acid increases expression of nicotinamide phosphoribosyltransferase, leading to increased NAD+ and downstream activation of histone deacetylase SIRT1 [168]. Maintaining low levels of pyruvic acid, through the upregulation of LDH-A to convert the metabolite into lactate, is one method by which cancers avoid apoptosis [169]. Changes in these metabolite levels as mice were put through withdrawal and re-exposure to both morphine and heroin indicate potential upregulation of compensatory pathways and underscore the impact of opioid exposure on TCA dysregulation and cancer-related pathways [97].

In an investigation performed on rat urine and serum samples, chronic morphine treatment (2–10 mg/kg/day for 5–14 days) increased metabolites including αKG, succinate, malic acid, and fumarate [81,96]. Interestingly, when rats were treated with both morphine and naloxone, many of the effected TCA metabolites remained elevated compared to control but reduced compared to the opioid-only treatment group, suggesting that metabolic changes brought on by opioid treatment may not entirely depend on opioid receptor activation [81]. αKG is also central to tumor growth dynamics as it is both a metabolite of the TCA, a key player in glutaminolysis, and acts as an epigenetic regulator through its role as co-factor for ten-eleven translocation enzymes and JmjC domain-containing histone demethylases [170]. In general, increased αKG induces anti-cancer effects [171]. The metabolite was shown to induce apoptosis through the ROS-PI3K/Akt/mTOR pathway, increase antineoplastic treatment efficacy through epigenetic upregulation of PD-L1, and reduce metastatic potential by suppression of TGF-β and VEGF production [162,172,173].

In brain tissue samples from morphine-dependant rats, changes to TCA metabolites were dependant on substructure localization. Succinate was increased in the nucleus accumbens but decreased in the striatum, while αKG was drastically decreased in only the hippocampus. Almost all observed changes to TCA metabolites disappeared with methadone co-treatment. It appears that chronic and consistent morphine treatment increases αKG in serum but decreases the compound in neuronal tissue. The combination of increased lactic acid, mentioned previously, and decreased succinate brought on by morphine treatment suggests a mitochondrial functional deficiency and decreased TCA cycle activity leading to decreased ATP levels with opiate exposure [90]. Proteomic changes across numerous studies in neuronal tissue following morphine treatment indicated changes to only one TCA enzyme, MD. The expression of this enzyme was reduced in rat brain tissue and monkey cerebral spinal fluid [83,98,99]. MD is often overexpressed in cancers and is associated with poor prognosis [174,175]. Cytosolic MD can act as a source of NAD required by cancer cells to maintain high levels of glycolysis [100,176]. Reduction in both cytosolic and mitochondrial MD expression as a result of morphine treatment may offer a potential strategy to mitigate the overexpression and enzyme hyperactivity observed in highly proliferative tumors.

A sweeping, definitive assertion regarding the consistent up- or downregulation of TCA cycle activity induced by opioids, specifically morphine, cannot be made. The complex and context-dependent nature of opioid-induced modulation of TCA activity necessitates careful consideration of multiple factors such as treatment protocols (including dosing amounts, frequency, and length), tissue types, anatomical substructure, and genetic heterogeneity. However, it can be concluded that opioid treatment leads to significant fluctuations of metabolites within the TCA cycle. These findings serve to emphasize the fundamental interconnectivity that characterizes cellular metabolism, whereby the TCA cycle acts as a central hub integrating diverse metabolic pathways highly influential in cancer cell growth and survival. Consequently, the intricate network of metabolic interactions becomes disrupted upon opioid exposure, leading to pronounced alterations in energy utilization and biosynthesis that may also occur in the context of tumor metabolism [177].

## 8. Oxidative Phosphorylation (OXPHOS)

### 8.1. OXPHOS in Cancer

Mitochondria are dynamic organelles consistently undergoing fusion and fission throughout the cell cycle. These organelles exist as either branched tubular networks, or fragmented granules depending on the cellular state, energy demand, and oxidative stress regulation [178]. Numerous studies have highlighted the imbalance of increased fission or decreased fusion in cancers leading to fragmented mitochondrial networks [179]. Enhanced mitochondrial fission has been shown to push cells into mitosis and increase replication [180]. Moreover, prevention of mitochondrial fragmentation through upregulation of the fusion protein Mfn1 or downregulation of the fission protein Drp1 significantly reduces metastatic abilities in certain cancers [181]. Cells with fragmented mitochondria have decreased OXPHOS capabilities and more heavily rely on glycolytic energy production [182,183]. The ETC is the most efficient metabolic pathway for the generation of ATP per unit of carbon source [184]. The pathway consists of five complexes, the first four of which generate a proton gradient across the inner mitochondrial membrane and thus produce the mitochondrial membrane potential (Figure 3). The fifth complex, ATP synthase, produces ATP while pumping protons back into the mitochondrial matrix [185]. Due to leaking electrons during this transport, superoxide and hydrogen peroxide formation occur at complex I and complex III [186,187]. The increase in ROS can promote the development and progression of tumors but must be balanced appropriately with detoxifying metabolic pathways to prevent ROS-activated apoptotic signaling [65,188].

Downregulation of mitochondrial metabolic enzymes, particularly those involved in OXPHOS, show a significant negative correlation with endothelial-to-mesenchymal transition (EMT) across over 20 types of cancers [189]. Metastatic cancers exhibit a profound upregulation of EMT in stark contrast to their parental tissue, underscoring the pivotal role of this molecular process in the acquisition of aggressive phenotypes [189,190]. An in-depth investigation into patient survival reveals an association between dysregulated EMT and the impairment of ETC enzyme expression wherein OXPHOS was discovered to be the most affect pathway among cohorts classified as low vs. high survival rates. ETC enzymes were found to be downregulated in 60% of cancer types. In samples with high induction of EMT and metastatic potential, OXPHOS was the most significantly downregulated metabolic pathway. The most frequently downregulated gene among low survival patients are subunits of complex I and IV [189]. The association between poor clinical outcome and downregulation of OXPHOS enzymes is attributed to the correlated increase in EMT. Targeting complexes I–IV of the ETC was shown to successfully reduce tumor growth and induce apoptosis [191,192,193]. Overall regulation of OXPHOS unsurprisingly varies in different cancer types and can even be heterogenous within tumors [194,195,196].

### 8.2. Opioid Modulation of OXPHOS

A study examining mitochondrial morphology following opioid treatment found that morphine had a minimal effect, while fentanyl and methadone significantly altered organelle morphology. Following four hours of treatment to murine neurons, methadone and fentanyl at concentrations between 25 and 100 μM caused a decrease in total mitochondrial area and branching [197]. A reduction in mitochondrial branching can be indicative of impaired ATP synthesis, disruption in ROS balance, and impaired calcium regulation [198]. As previously mentioned, mitochondrial structure and function are integrally related. Increased fragmentation of mitochondria, such as that induced by fentanyl treatment, can lead to higher levels of cancer cell proliferation and metastasis [180,181]. No significant changes to complex I or III activity in mouse brain tissue occurred after 14 days of treatment with morphine at high concentrations (10–40 mg/kg) [86], but the expression levels of complex I and III were decreased in the amygdala of acutely treated mice [93]. A reduction in complex I expression was found in the hippocampus of rats following chronic morphine treatment [87]. Morphine stimulates nitric oxide (NO) release from the mitochondria in many cell types [199,200,201]. NO can interact with and inhibit complex IV of the ETC, impeding cellular respiration [202]. Morphine treatment of glioma cells resulted in a 30% decrease in the mitochondrial membrane potential via activation of NO-release pathways. The observed effect was believed to be a result of complex IV inhibition but interestingly was unique to morphine. Fentanyl and methadone do not substantially increase NO release, again indicating the off-target effects of morphine [203]. Mitochondrial functional analysis (oxygen consumption) in isolated rat liver mitochondria after incubation with morphine, cocaine, and a combination of the two, indicated that morphine treatment decreases complex I activity by about 20%. No significant changes to complex I or II activity were observed in neuronal mitochondria in this study. All three drug groups did result in decreased ATP synthase activity, however, no changes to mitochondrial membrane potential were observed [204]. In rhesus monkeys treated with 3–15 mg/kg of morphine for 7–20 days, ATP synthase expression was decreased two-fold [83,205]. A similar conclusion was drawn from morphine exposure in rats which identified a two-fold decrease in ATP synthase activity following an acute treatment period of three days [98,106]. This supports similar findings of decrease in ATP metabolite levels following morphine treatment [87,138]. Investigation of complex II–IV expression in a variety of murine and primate models predominantly indicate no change following morphine treatment; however, studies demonstrate either a decrease in activity or expression of both complex I and ATP synthase [83,85,87,93,98,106,204,205,206,207]. As previously mentioned, outside of glycolysis, ATP synthase is a large source of ATP for cancer cells. In clear cell renal cell carcinoma, glioblastomas, and HER2+ breast tumors, components of ATP synthase are often overexpressed [208,209,210]. Several drugs targeting the activity of ATP synthase, both synthetically designed (e.g., bedaquiline) and naturally occurring (e.g., curcumin; NCT03769766; NCT02439385) are currently under preclinical or early stage clinical trial investigation for amelioration of current antineoplastic agents [141,211,212]. Reduction in ATP levels and ATP synthase activity as a result of opioid treatment may synergistically work with antineoplastic agents much like the aforementioned drugs undergoing clinical trials.

In human hepatoma cells, in vitro fentanyl treatment did not result in significant changes to cellular respiration, the activity of any components of the ETC, or an increase in intracellular ROS levels. Despite decreased ATP levels, fentanyl did not impact ATP synthase activity [213]. Interestingly in non-cancerous rat brain samples, fentanyl did impact mitochondrial bioenergetics. Moderate concentrations of fentanyl were able to inhibit complexes II to IV of the ETC, while higher concentrations inhibited ATP synthase by over 50% [214]. Similarly, 30-day tramadol treatment in mice resulted in significant inhibition of complex I, III, and IV in isolated mitochondria from brain samples [215].

The majority of oxidative metabolism analysis in the context of opioids is studied in brain tissue samples since mitochondrial glucose oxidation is the main energy source in neurons and the brain is the primary clinical opioid effect site [216]. While drug concentration and treatment model vary from study to study, the literature supports the idea that opioids reduce OXPHOS capacity, particularly in neuronal tissue. Numerous studies have recently indicated that inhibition of OXPHOS can be a strategy to overcome chemotherapeutic resistance [217,218,219]. Other OXPHOS inhibitors, such as IACS-010759, have been shown to strongly inhibit proliferation [220]. Although not all cancers heavily rely on OXPHOS for energy production, the concomitant administration of opioids alongside antineoplastic interventions may confer additional advantages for those cancer subtypes exhibiting energy dependence.

**Figure 3 pharmaceutics-15-02225-f003:**
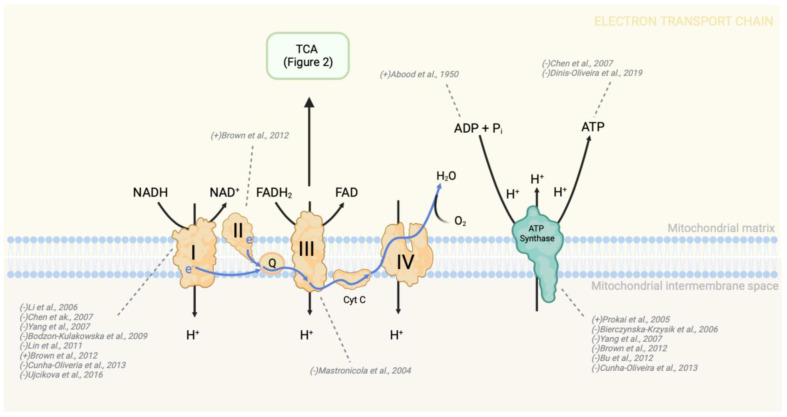
Changes in OXPHOS enzyme activity due to morphine treatment. Morphine treatment consistently leads to a decrease in the activity of complex I and ATP synthase. Consequently, a reduction in intercellular ATP levels is expected following morphine treatment. To indicate the directionality of change in metabolite or protein levels, brackets preceding the study authorship are used. In this context, a “+” symbol denotes an increase in activity, expression, or level, while a “-” symbol indicates a decrease in the aforementioned parameters. NADH—nicotinamide adenine dinucleotide (reduced form), FADH2—flavin adenine dinucleotide (reduced form); Complex I—NADH dehydrogenase; Q—coenzyme Q (also known as ubiquinone); Complex II—succinate dehydrogenase; Complex III—cytochrome bc1 complex; Cyt c—cytochrome c; Complex IV—cytochrome c oxidase [80,83,85,87,93,98,106,137,203,204,205,206,207].

## 9. Metabolic Mitigation of Oxidative Stress

### 9.1. Metabolic Mitigation of Oxidative Stress in Cancer

With elevated levels of metabolism and cellular proliferation, cancer cells have higher quantities of ROS compared to non-cancer cells [221,222]. While increased ROS can act as a pro-oncogenic signaling pathway, they can also trigger apoptosis, autophagy, and other forms of cell death [223,224]. As such, many cancers upregulate compensatory pathways such as the PPP, MD-1, malic enzyme-1, isocitrate dehydrogenase isoenzymes, as well as serine metabolism, to increase NADPH production to be used for ROS detoxification by the GSH cycle (Figure 4) [225]. The PPP branches from glycolysis at the node of G6P formation to generate phosphopentoses and ribonucleotides for the production of nucleic acids. It is also a large supplier of NADPH, required for reduction of GSH and scavenging of ROS, showing involvement in both redox balance and the biosynthesis of macromolecules [226].The pathway has an oxidative and non-oxidative phase that can be preferentially upregulated allosterically via the pathway’s own products depending on the demand of the cell. Each G6P molecule made during glycolysis can be funneled into the oxidative phase of the PPP for the formation of two NADPH molecules and one phosphorylated pentose ring. The non-oxidative phase of the pathway can then produce glyceraldehyde-3-phosphate and fructose-6-phosphate, which may be fed back into glycolysis. NADPH produced from the PPP can be used in combination with acetyl-coA by fatty acid synthases for fatty acid production [227,228]. More importantly, in the context of cancer, NADPH is used by the GSH cycle for detoxification of peroxide radicals. In many cancers, the high consumption of NADPH for ROS mitigation results in upregulation of expression or activation of G6PDH, the first PPP enzyme that produces NADPH and also the rate limiting step of the pathway [229]. Similarly, artificial downregulation or mutation of G6PDH increases cellular oxidative stress, ROS accumulation, and leads to increased apoptosis [230,231]. Glutathione reductase (GR) drives the NADPH-dependent reduction of glutathione disulfide (GSSG) to GSH. In turn, GSH can then be oxidized back into GSSG by members of the glutathione peroxidase (GPx) family as they convert organic hydroperoxides into corresponding alcohol or H_2_O_2_ into water [232]. Elevated levels of GSH have been found in a variety of tumors including ovarian, breast, and lung cancers and has also been suggested as necessary for proliferation and tumor initiation [233,234,235]. In other tumor types, such as brain and gastrointestinal tumors, GSH reduction is observed [235]. This heterogeneity is predominantly attributed to the varying levels of GSH regulation by the healthy tissue [235]. Though some studies present a relationship between high GSH levels and poor patient outcome, others do not support such findings [236,237,238,239].

### 9.2. Opioid Modulation of Oxidative Stress Mitigation Pathways

G6PDH is the first enzyme and rate-limiting step of the PPP [240]. Enzyme kinetic investigations on G6PDH isolated from healthy human erythrocytes indicated that morphine is a weak non-competitive inhibitor of G6PDH with a K_i_ of 25.93 ± 6.48 mM and IC50 of 43.58 mM. The study found that vancomycin, a glycopeptide antibiotic, and omeprazole, a proton pump inhibitor, were also non-competitive inhibitors of G6PDH. The effect of ketamine and remifentanil were also partially investigated and despite showing some inhibitory effects, the IC50 values of these anesthetic agents were sufficiently higher than morphine, indicating less inhibition of G6PDH [241]. Contrary to changes in activity level, expression of G6PDH does not appear to change following chronic morphine treatment (10 mg/kg for 5 days in isolated rat amygdala) [93]. As previously mentioned, acute morphine treatment at high doses of 50 mg/kg for fewer than 2 h in rats decreased G6P in the brain while lower concentrations (15 mg/kg) increased G6P [80]. Unfortunately, there are insufficient data on the effect of morphine or other opioids on direct changes to NADPH, the product of G6PDH. While the PPP is a major source of NADPH, other enzymes, such as isocitrate dehydrogenase-1 (IDH1) and MD, can also produce this molecule [242]. G6PDH upregulation is highly observed in a variety of cancers including breast carcinoma, glioma, and lung adenocarcinoma. This upregulation is strongly correlated with poor patient prognosis through perpetuation of aggressive cancer phenotypes such as increased growth, invasion, migration, and chemotherapy resistance [230,243,244,245]. It is therefore not surprising that inhibitors of G6PDH and transcriptional regulators are being investigated as therapeutic options. Polydatin, a glucoside of resveratrol and inhibitor of G6PDH, was shown to impede cell proliferation and block transition into S-phase [246]. In cancers that are highly sensitive to G6PDH activity, enzymatic inhibition by opioids could have a beneficial effect similar to that of polydatin. 

Anesthetic drugs, midazolam and propofol, were found to significantly upregulate numerous PPP metabolites in cerebral spinal fluid of subarachnoid hemorrhage patients. Samples taken up to 72 h post sedation indicated that midazolam led to increased 6-phosphogluconate, ribulose-5-phosphate, erythrose-4-phosphate, and glyceraldehye-3-phosphate, while propofol increased ribose-5-phosphate, erythrose-4-phosphate, glyceraldehyde-3-phosphate, and fructose-6-phosphate. Conversely, dexmedetomidine slightly inhibited the PPP, resulting in decreased NADP and sedoheptulose-7-phosphate [247]. Given the neuroprotective and antioxidant properties associated with both propofol and midazolam in the brain, the increase in PPP activity is likely not in response to ROS accumulation or upregulation of pathways to counteract oxidative damage brought on by anesthetic treatment [248,249,250]. Instead, upregulation of the PPP is thought to occur to promote biosynthesis given that purine and pyrimidine metabolism were both also upregulated in cerebral spinal fluid samples. The observed phenomenon was thought by the authors to represent a source of “rapid sedative induced DNA synthesis”.

Much like an increase in G6PDH expression and activity, upregulation of the PPP, in particular for the purpose of biosynthesis, supports the biomaterial demand required of cancer cells to maintain high levels of proliferation. Despite being beneficial for its neuroprotective properties, propofol has also been shown to transcriptionally activate Nrf2 pathways involved in enhanced expression of ROS detoxification enzymes, and thus increase proliferation and invasion of gallbladder and breast cancer cells [251,252]. Propofol-induced upregulation of the PPP is likely a metabolic pathway through which the anesthetic agent may promote metastatic phenotypes.

A predominant utility for PPP-produced NADPH is for the mitigation of ROS through antioxidant pathways, most notably GSH metabolism [253]. Morphine treatment decreases GSH metabolite levels in murine models and human samples [254,255,256,257]. Following a single intraperitoneal dose of morphine at 3, 6, or 12 mg/kg in adult mice, GSH levels were significantly reduced in brain samples [257]. In a different study investigating the attenuation of morphine tolerance in mice by inhibition of nitric oxide synthase (NOS), subcutaneous morphine injections of 5 mg/kg twice a day resulted in a significant decrease in GSH and GSH peroxidase activity in brain samples. Changes to both the metabolite level and enzyme activity were first detected at 3 days but were sustained for the full 7-day experiment [255]. A similar drop in GSH and GSH peroxidase and an increase in ROS was observed in primary hippocampus cultures prepared from newborn mice following a 24-h incubation with 1 μM morphine. However, pre-treatment with L-NAME, a NOS inhibitor, increased GSH concentrations and reduced ROS levels relative to morphine-only treatment, suggesting that inhibition of the NOS system plays a key role in the oxidative stress response of the hippocampus [256]. Along with the ability for NADPH to attenuate ROS via GSH metabolism, it can also act as a substrate for NOX enzymes, also known as NADPH-oxidases [258]. This is of particular importance in the context of both cancer and opioid metabolism given that morphine has been shown to activate NOX enzymes which consume NADPH for superoxide production [259,260,261]. According to the existing literature, morphine induces a global reduction in PPP activity, accompanied by an elevation in NADPH-consuming enzymes, thereby compromising cancer cells’ ability to counteract ROS and reducing the availability of nucleotides necessary for DNA synthesis and proliferation. Conversely, other anesthetic agents appear to upregulate the PPP. However, the majority of investigations into PPP upregulation by anesthetic agents are primarily conducted within the framework of intraoperative neuroprotection as opposed to directly investigating changes in cancer cells.

**Figure 4 pharmaceutics-15-02225-f004:**
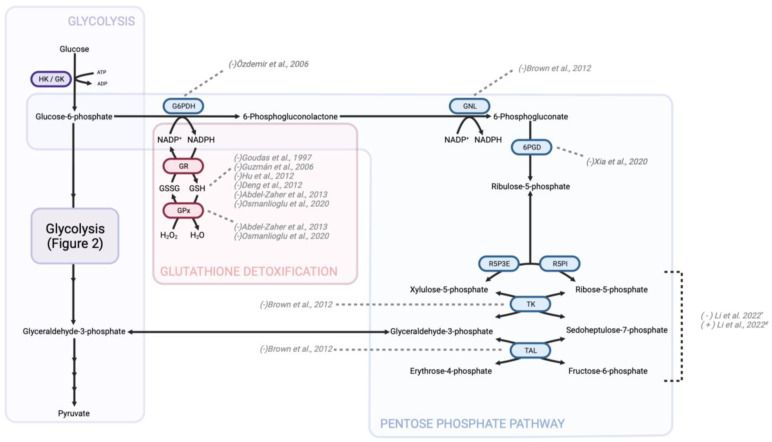
Changes to GSH detoxification and the PPP secondary to opioid treatment. GSH detoxification is a vital cellular process involved in the neutralization of reactive oxygen compounds and maintenance of redox balance. Opioids were found to impact this pathway by influencing GSH levels and the activity of GSH-related enzymes, such as GPx and glutathione-S-transferase. Studies suggest that chronic opioid use can lead to a decrease in GSH levels and impair the efficiency of GSH detoxification, potentially contributing to oxidative stress and cellular damage. To indicate the directionality of change in metabolite or protein levels, brackets preceding the study authorship are used. The accompanying figure illustrates the alterations induced by morphine treatment, with the exception of those marked with a “#” signifying changes attributable to midazolam and propofol, or a “*” signifying changes resulting from dexmedetomidine administration. HK—hexokinase; GK—glucokinase; G6PDH—glucose-6-phosphate dehydrogenase; GSH—glutathione (reduced form); GSSG—glutathione disulfide (oxidized form); GPx—glutathione peroxidase; GR—glutathione reductase; NADPH—nicotinamide adenine dinucleotide phosphate (reduced form); NADP+—nicotinamide adenine dinucleotide phosphate (oxidized form); GNL—gluconolactonase; 6PGD—6-Phosphogluconate dehydrogenase; R5PI—ribose-5-phosphate isomerase; R5P3E—ribose-5-phosphate-3-epimerase; TK—transketolase; TAL—transaldolase [83,90,91,92,241,247,254,255,256,257].

## 10. Conclusions

Opioid use for pain management is essential and ubiquitous for patients with cancer. While effective for reducing nociception, opioids, regardless of class, have off-target interactions which lead to predominantly unwanted side effects [262]. Retrospective and cohort studies suggest that opioid use is predominantly associated with cancer recurrence, though this observation is nuanced and dependent on tissue specificity and genomic alteration [43,263]. Unfortunately, investigations into the impact of opioids on tumorigenesis with data specific to metabolic changes are not widely available [19]. In this review, we have focused on metabolic changes brought on by opioid exposure in a variety of tissue sources. Research in this field is limited by the diversity in sample origin, treatment model, and choice of opioid [137]. Observations from pre-clinical in vivo models treated with morphine specifically focusing on biomarkers of addiction and withdrawal dominate the data pool, while limited research exists on the effects of other opioids like fentanyl, methadone, oxycodone, and tramadol. More surprisingly, no reliable data can be found on other commonly used opioids in relation to cellular metabolism, such as hydromorphone or buprenorphine. Given the fundamental importance of metabolic reprogramming in cancer cells, any changes to key metabolic pathways brought on by external factors, such as pain management options, must be considered to tailor treatment and improve patient prognosis through precision medicine. Cancer patients are often treated with opioids for long durations and receive a cocktail of medications for disease management [5,14]. The profound impact of cellular metabolism reprogramming on cancer growth and chemotherapeutic resistance is widely recognized. The amassed data presented above unmistakably demonstrate that opioids exert influence on cellular metabolism and redox equilibrium. However, better comprehending the mechanisms, extents, and shifts in metabolic pathways within distinct cancer types is pivotal. Such understanding could profoundly influence decisions regarding pain management strategies, potentially affecting disease progression and patient outcomes. Apart from fundamental similarities in proliferation, cancer is a heterogenous disease with hundreds of further classifications by which tumors can be differentiated [62]. This includes genetic mutations, epigenetic signatures, and distinct changes in metabolism—as discussed above [264,265,266,267].

Advocacy for precision medicine among the cancer patient population has increased [268]. Given the outlined opioid-induced changes to cellular metabolism and the diversity in tumor metabolic reprograming, it is likely that the choice of pain management can play a larger role in disease treatment and response than previously appreciated. Patient heterogeneity must be leveraged by data-driven methodology to inform treatment decisions such that the correct patient receives the appropriately tailored treatment based on their individual characteristics [269]. A nuanced understanding of these complexities is essential for unraveling the broader impact of opioids on cellular metabolism and may contribute to the development of targeted therapeutic interventions.

## Figures and Tables

**Figure 2 pharmaceutics-15-02225-f002:**
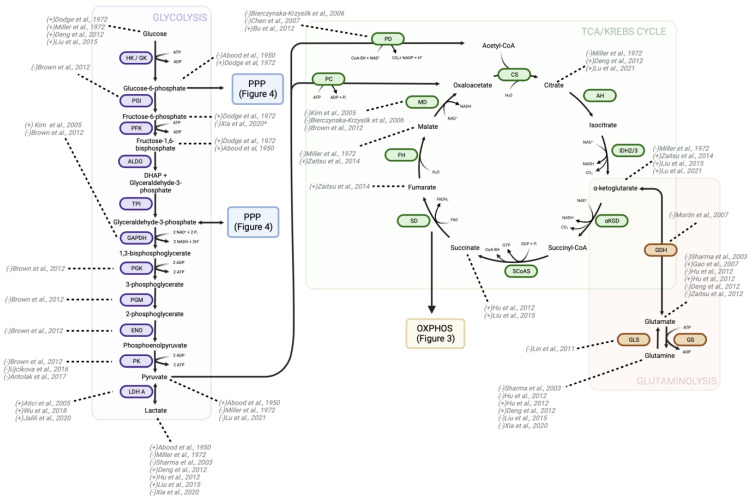
Changes in metabolites and enzymes of glycolysis, TCA, and glutaminolysis due to morphine treatment. There is consensus among studies regarding increased glucose transport into the cell, upregulation of LDH-A, and decreased expression/activity of PK, but the remaining steps of glycolysis display variable patterns of upregulation and downregulation by morphine. Three separate studies have found that morphine decreases malate dehydrogenase expression; however, the change in other enzymes and metabolites of the TCA vary within the literature. Glutamine metabolism, or glutaminolysis, directly feeds into the TCA through α-ketoglutarate production. Morphine and other opioids affect glutaminolysis metabolites and enzymes, yet conflicting findings exist about whether they amplify or diminish this metabolic pathway. Directionality of change in metabolite or protein is indicated in the brackets preceding study authorship. “+” indicates an increase in either activity, expression, or level, while “-” indicates a decrease in the aforementioned parameters. HK—hexokinase; GK—glucokinase; PGI—phosphoglucose isomerase; PFK—phosphofructokinase; ALDO—aldolase; DHAP—dihydroxyacetone phosphate; TPI—triosephosphate isomerase; GAPDH—glyceraldehyde-3-phosphate dehydrogenase; PGK—phosphoglycerate kinase; PGM—phosphoglycerate mutase; ENO—enolase; PK—pyruvate kinase; LDH-A—lactate dehydrogenase A; PD—pyruvate dehydrogenase; PC—pyruvate carboxylase; CS—citrate synthase; AH—aconitase or aconitate hydratase; IDH—isocitrate dehydrogenase; a-KGD—α-ketoglutarate dehydrogenase; SCoAS—succinyl CoA synthetase; SD—succinate dehydrogenase; FH—fumarase or fumarate hydratase; MD—malate dehydrogenase; GLS—glutaminase; GS—glutamine synthase; GDH—glutamate dehydrogenase (also known as glutamic acid decarboxylase); OXPHOS—oxidative phosphorylation; PPP—pentose phosphate pathway [18,79,80,81,82,82,83,84,85,86,87,88,89,90,91,91,92,92,93,94,95,96,97,98,99,100].

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
