# Peer review of "Impact of Opioids on Cellular Metabolism: Implications for Metabolic Pathways Involved in Cancer"

_pharmaceutics, 2023, doi:10.3390/pharmaceutics15092225_

Round 1

Reviewer 1 Report

The text „Impact of Opioids on Cellular Metabolism: Implications for Metabolic Pathways involved in Cancer” offers interesting input on wider consequences, or possible wider consequences, of morphine and other opioids in cancer treatment. The possible interactions of opioids and cancer cells are of great importance, as the Authors presented in the conclusions.

However, the cautious comments on the possibility of secondary effects are located at the end of the text, which may lead to early assumptions on dangers of pain management in cancer treatment. The statements in the text are in most cases guarded, with only presumed or suggested effects, although selective reading or just web search results may cause problems and stress in patients and practitioners. In the case of high impact open access journals, the effect of provided information may be higher than usual.

In my opinion, an additional cautious statement should be included in the abstract and introductory part, on the diversity of cancer types, specificity of data and limited amount of direct experiments (elements could be already found, my suggestion is directed at stronger warning). At the same time, the necessity of more detailed studies is worth underscoring.

For example, fragment 337-345 ends with treatment suggestion, but the provided data are for animal models, whereas the described logical path is attractive, but speculative. The statement (460) “Morphine may differentially impact cancer progression based on tumor location, possibly increasing aggression of hippocampal or CNS tumors.” is based on previously discussed data, and is cautious and speculative, however, it could be taken out of context and used as treatment suggestion – or malpractice.

The Authors prepared a text that is full of information, with 266 references. The division into Glycolysis, Creatine metabolism, Glutamine metabolism, TCA cycle, Oxidative Phosphorylation, and within these chapters, into description of the process, cancer relation and opioid effects is well planned. Separating paragraphs would help in reading, as at the moment the text is not easy to use. Some substances and pathways (aKG, PPP) are mentioned in more than one place, with different references. Corelating these statements could be useful

The figures are very interesting and the idea of presenting the +/- effects described in references is excellent. If it would be possible to add reference numbers to the cited author`s names, finding the indicated references will be much easier.

The number of abbreviations is really high, as expected in a paper related to metabolic pathways. Some seem to be missing the full version, or they occur in much further parts of the text, separated from explanation, complicating the reading.

Greek letters seem affected by PDF preparation? (alpha in TNF and a-ketoglutarate, kappa in NFkB, possibly micro-ultra uM), italics in “in vivo and in vitro”

Minor questions and requests for clarification:

Line 94: NSCLC patients who had experienced tumor recurrence in a 5-year time period had received 2 times the amount of postoperative morphine administration[42] – does this concern the frequency of administration, dose or number of patients?

Line 108: is there any possibility that the effect is related to the lack of gastric problems caused by morphine, resulting in better nutrition and quality of life effect?

Line 116: In models of breast cancer carcinoma and lung cancers?

Figure 1: the caption is quite extensive, maybe transferring part of it to the main text would be beneficial, as well as locating the figure earlier in the paragraph.

In Glutamine Metabolism (6.1), there are two sentences describing glutamine addiction, with various cancers/tumors. Please consider rewriting.

Line 425: comparator drug?

Line 436: histone deacetylace?

Line 442: chronic morphine treatment (2-10mg/kg/day for 5-14 days) increased including aKG, succinate, malic acid, and fumarate…

Line 531: Please verify the unit (ultramolar?): at concentrations between 25-100 uM

Line 648: please verify the value: G6PDH with a Ki of 25.936.48 mM

Line 760: Please clarify: understanding how, to what degree, and the directionally of change of key metabolic pathways in specific cancer types may significantly alter disease progression and patient outcome

The language of the text is of high quality, which, surprisingly, may cause problems. Some phrases are quite advanced and may affect the reading and understanding of non-native speakers (examples: (81) numerous studies amalgamate the results, (712): impeding the availability of nucleotides necessary for DNA synthesis and proliferation)

There are places in the text that explanation is needed:

Line 65: is the main cite of action for opiates

Line 68: Understanding these changes will better inform decisions on opioid prescription and use for cancer patients,

Line 114: Numerous comprehensive reviews have highlighted that opioids, particularly morphine, have divergent effects of cancer phenotypes

Line 198: differential affects

Line 478: anatomica substructure?

Line 564: components of ATP synthase is often overexpressed

Line 744: Unfortunately, investigations into the impact of these drugs on tumorigenesis with data specific to metabolic changes is not widely

Line 763: cancer is a heterogenous diseases?

Line 766: Aadvocacy

Reference 266 – data missing?

There are parts of the text that certain problems appear; singular/plural verb, affect/effect

Reviewer 2 Report

This review article covers the influence of common opioids on cell metabolism and cancer diseases. It is concise, well written, and interesting to read. The provided figures are clear and informative. But there are some stylistic issues and typo errors. Thus, I recommend minor revision of the manuscript:

Introduction: Please also provide a figure or scheme with the chemical structures of the opioids mentioned in the manuscript and/or in Table 1.

The non-addictive opium alkaloid noscapine is a thoroughly investigated anticancer active natural compound, and although it does not target opioid receptors, noscapine might be briefly mentioned and/or discussed in this review manuscript in the light of the presented data of common opioids. In particular, noscapine suppressed the Warburg effect in colon cancer cells (PMID 32606759).

Figures 1-3: Please also provide the numbers of the shown references.

Please check again the abbreviation F-1,6-B. I think F-1,6-BP is more common.

Please use the Greek letters in NFkB, TNF-a, TGF-b and aKG.

Line 196: Please correct ´´… -phospate´´. The same in line 265.

Line 230: Please replace ´´Myc-c´´ by a more common quotation.

Line 357: Please correct ´´additctet´´.

Line 407: I suggest ´´The TCA cycle is a central hub …´´.

Line 500: Replace ´´relay´´ by ´´rely´´.

Lines 672 and 674: Please correct phoasphate, glyceraldhye, and glyceraldehye.

References: There are some inconsistencies in this section such as names and texts in big letters (e.g., ´´Beckett´´, ´´WARBURG´´ and the compete title of reference 31). In addition, the journal name and page numbers are missing in ref 31. Please correct according to the journal guidelines.

The text is fine, there are only some typographical errors in the manuscript.
